# Comparison of Ablation Area and Change in Functional Liver Reserve after Radiofrequency Ablation for Hepatocellular Carcinoma Using the arfa^®^ and VIVA^®^ Systems

**DOI:** 10.3390/jcm11020434

**Published:** 2022-01-15

**Authors:** Hiroaki Takaya, Tadashi Namisaki, Kazusuke Matsumoto, Junya Suzuki, Koji Murata, Yuki Tsuji, Keisuke Nakanishi, Kosuke Kaji, Mitsuteru Kitade, Ryuichi Noguchi, Hitoshi Yoshiji

**Affiliations:** Department of Gastroenterology, Nara Medical University, Shijo-cho 840, Kashihara 634-8522, Japan; tadashin@naramed-u.ac.jp (T.N.); discovery97mac@docomo.ne.jp (K.M.); suzukij@naramed-u.ac.jp (J.S.); muratak@naramed-u.ac.jp (K.M.); tsujih@naramed-u.ac.jp (Y.T.); k-zo@drive.ocn.ne.jp (K.N.); kajik@naramed-u.ac.jp (K.K.); kitademitsu@yahoo.co.jp (M.K.); rnoguchi@naramed-u.ac.jp (R.N.); yoshijih@naramed-u.ac.jp (H.Y.)

**Keywords:** hepatocellular carcinoma, radiofrequency ablation, adjustable electrode needle, ablation area, nonalcoholic steatohepatitis

## Abstract

Radiofrequency ablation (RFA) is recommended in Japan for patients with hepatocellular carcinomas (HCCs) one to three in number and ≤3 cm in size. The arfa^®^ and VIVA^®^ RFA systems are widely used for patients with HCC and this retrospective observational study aims to compare their performances. The study included 365 patients with HCCs one to three in number and ≤3 cm in size who underwent RFA using the arfa^®^ system (arfa^®^ group) or the VIVA^®^ system (VIVA^®^ group). The total bilirubin (T-Bil) level after RFA was higher in the arfa^®^ group than in the VIVA^®^ group. With a 3-cm electrode needle, the longest diameter (Dmax) and the shortest diameter were analyzed and found to be greater in the arfa^®^ group than in the VIVA^®^ group. Furthermore, Dmax with the 2.5-cm electrode needle was greater in the arfa^®^ group than in the VIVA^®^ group. Statistically significant differences in the ablation area and in the T-Bil value after RFA were observed between the groups; however, these differences are not considered clinical problems because the difference in the ablation area was only slight and the Child–Pugh score was the same between the groups. Thus, hepatologists can use either of the RFA systems based on their preference.

## 1. Introduction

Hepatocellular carcinoma (HCC) is the fourth leading cause of cancer-related death in the world [1,2,3]. The medical treatment policy in Japan is based on the consensus-based Clinical Practice Guidelines for HCC Management proposed by the Japan Society of Hepatology (JSH) [4]. Radiofrequency ablation (RFA) or surgical resection is recommended for patients with HCCs one to three in number and ≤3 cm in size [4]. HCC is mainly caused by hepatitis C virus-induced liver cirrhosis [5,6,7]. However, other causes include hepatitis B virus, excessive alcohol consumption, nonalcoholic steatohepatitis, autoimmune hepatitis, and primary biliary cholangitis. In previous studies, it was reported that there is no difference in prognosis between patients with HCC who undergo RFA and patients with HCC who undergo surgical resection [8,9,10,11]. RFA is less invasive than surgical resection, and it can be used for elderly patients with reduced performance status and for patients with liver cirrhosis and reduced functional liver reserve.

In recent times, adjustable electrode needles are widely used for RFA in Japan [12]. Healthcare cost in Japan is on the rise and needs to be controlled [13], and in terms of cost, adjustable electrode needles are effective as they can be used for HCCs of different sizes. The arfa^®^ (Japan Lifeline Co. Ltd., Tokyo, Japan) and VIVA^®^ (STARmed Co. Ltd., Goyang, Korea) RFA systems with adjustable electrode needle are commonly used in Japan for RFA of patients with HCC. Due to the high risk of recurrence of HCC, it is important to ensure that the ablation area is sufficient. Further, since liver cirrhosis is the cause of HCC, it is essential that decline in functional liver reserve after RFA is mild. Moreover, it is unclear if a difference in performance exists between the arfa^®^ and VIVA^®^ RFA systems.

In this study, we compared arfa^®^ and VIVA^®^ RFA systems with respect to the ablation area and functional liver reserve in patients with HCC.

## 2. Materials and Methods

### 2.1. Patients and Study Design

This study was a retrospective observational study that included 409 patients with HCC who underwent RFA at our hospital between April 2016 and November 2021. HCC was diagnosed using dynamic computed tomography (CT) and/or dynamic magnetic resonance imaging (MRI) in accordance with the consensus-based Clinical Practice Guidelines for HCC Management proposed by the JSH [4]. The eligibility criteria were HCC tumor number of one to three and HCC tumor size ≤ 3 cm. The exclusion criterion was patients with HCC who underwent combined therapy (e.g., RFA and transcatheter arterial chemoembolization or RFA and percutaneous ethanol injection therapy) [14,15]. The other exclusion criterion was patients with HCC who required additional RFA. For these patients, complete ablation was not achieved with one RFA session; therefore, an additional RFA session was performed a few days later. A final total of 365 patients with HCC were included in our study. All the patients were in Child–Pugh class A or B and did not have extrahepatic metastasis or vascular invasion [16]. First, we investigated change in functional liver reserve before RFA and one day after RFA in patients who underwent RFA using the arfa^®^ system (arfa^®^ group) and those who underwent RFA using the VIVA^®^ system (VIVA^®^ group). Next, we assessed the ablation area of the arfa^®^ system and that of the VIVA^®^ system for each electrode needle length (Figure 1). This study was approved by the local ethics committee of Nara Medical University and was conducted in accordance with the ethical standards of the Declaration of Helsinki. All participants provided informed consent.

### 2.2. RFA

RFA was performed using the arfa^®^ or VIVA^®^ RFA system with adjustable electrode needle. We used the VIVA^®^ system from April 2016 to September 2020 and the arfa^®^ system from October 2020 to November 2021. All RFA sessions were performed via percutaneous approach under ultrasound guidance (LOGIQ E9 XDclear 2.0, GE Healthcare, Chicago, IL, USA). During RFA, intravenous conscious sedation was performed, and vital signs were monitored. All protocols were started with RFA at 40 W (2-cm 17G adjustable electrode needle), 50 W (2.5-cm 17G adjustable electrode needle), or 60 W (3-cm 17G adjustable electrode needle). The output of the VIVA^®^ system was increased at a rate of 10 W/min (auto mode) until the so-called “roll-off” or “break” (i.e., stop of delivery of RFA system) was achieved. In contrast, the output of the arfa^®^ system was increased at a rate of 1 W/6 s (linear mode) until the so-called “roll-off” or “break” (i.e., stop of delivery of RFA system) was attained. This protocol was repeated three times for the RFA of one HCC.

### 2.3. Follow-Up

Each patient underwent blood test and dynamic CT or dynamic MRI one day after RFA. We evaluated functional liver reserve before RFA and one day after RFA, and we determined the longest diameter (Dmax) and the shortest diameter (Dmin) of the ablation area.

### 2.4. Statistical Analysis

All statistical analyses were performed using EZR (version 1.54; Saitama Medical Center, Jichi Medical University), which is a graphical user interface for R (version 4.0.3; R Foundation for Statistical Computing, https://www.r-project.org, accessed on 10 January 2022). EZR is a modified version of R commander version 2.7-1 that includes statistical functions frequently used in biostatistics [17]. Results were expressed as medians and interquartile ranges. Differences between the groups were analyzed using Mann–Whitney U test, categorical data were analyzed using Fisher’s exact test, and a two-tailed *p*-value < 0.05 was considered significant.

## 3. Results

### 3.1. Patient Characteristics

Table 1 shows the patient characteristics. The median age of patients with HCC who underwent RFA was 75 (68–80) years. The study population consisted of 259 men and 106 women. Of these 365 patients, 75 had hepatitis B virus, 144 had hepatitis C virus, 58 abused alcohol, 64 had nonalcoholic steatohepatitis, 11 had autoimmune hepatitis, 4 had primary biliary cholangitis, and 9 had a cryptogenic condition. A statistically significant difference in the etiology of HCC was observed between the arfa^®^ and VIVA^®^ groups (*p* < 0.05). Albumin (Alb) level and prothrombin time (PT) were significantly higher in the arfa^®^ group than in the VIVA^®^ group (*p* < 0.05 for both variables). In addition, tumor size was significantly greater in the arfa^®^ group than in the VIVA^®^ group (*p* < 0.05).

### 3.2. Change in Functional Liver Reserve

We evaluated the change in functional liver reserve before RFA and one day after RFA. The absolute value of all liver parameters showed statistically significant changes before and 1 day after RFA (Figure 2). Between the arfa^®^ and VIVA^®^ groups, only the ratio of total bilirubin (T-Bil) level 1 day after RFA to that before RFA was statistically significant (Table 2).

### 3.3. Comparison of Ablation Area

The total number of HCCs in the patients was 438. Of these, we excluded 163 HCCs because they were punctured multiple times during RFA. Considering that only a few HCCs were treated using a 1.5 or 1-cm 17G adjustable electrode needle, we did not evaluate the ablation area in these HCCs. We determined the Dmax and Dmin of the ablation area for 115 HCCs treated with RFA using a 3-cm 17G adjustable electrode needle, 95 HCCs treated with RFA using a 2.5-cm 17G adjustable electrode needle, and 45 HCCs treated with RFA using a 2-cm 17G adjustable electrode needle that were punctured once in one RFA session. With the 3-cm electrode needle, Dmax and Dmin were greater in the arfa^®^ group than in the VIVA^®^ group (*p* < 0.05 for both variables) (Figure 3a,b), and no statistically significant difference was observed in the ratio of Dmin to Dmax (Dmin/max) between the arfa^®^ and VIVA^®^ groups (Figure 3c). With the 2.5-cm electrode needle, Dmax was found to be greater in the arfa^®^ group than in the VIVA^®^ group (*p* < 0.05) (Figure 3d). In contrast, no significant difference in Dmin was observed with the 2.5-cm electrode needle between the arfa^®^ and VIVA^®^ groups (Figure 3e). Further, with the 2.5-cm electrode needle, Dmin/Dmax was lower in the arfa^®^ group than in the VIVA^®^ group (*p* < 0.05) (Figure 3f). No difference in Dmax, Dmin, or Dmin/Dmax was observed with the 2-cm electrode needle between the arfa^®^ and VIVA^®^ groups (Figure 3g–i).

## 4. Discussion

In this study, we evaluated patients with HCC who underwent RFA in our hospital between April 2016 and November 2021. The VIVA^®^ RFA system was used from April 2016 to September 2020, and the arfa^®^ RFA system was used from October 2020 to November 2021. Patients in the VIVA^®^ group and patients in the arfa^®^ group were compared. A significant difference in the etiology of liver cirrhosis was observed between the two groups. In the VIVA^®^ group, the etiology of liver cirrhosis was hepatitis B virus in 23% of patients, hepatitis C virus in 42% of patients, alcohol abuse in 15% of patients, nonalcoholic steatohepatitis in 13% of patients, autoimmune hepatitis in 3% of patients, primary biliary cholangitis in 1% of patients, and cryptogenic condition in 2% of patients. In contrast, the etiology of liver cirrhosis in the arfa^®^ group was hepatitis B virus in 14% of patients, hepatitis C virus in 33% of patients, alcohol abuse in 18% of patients, nonalcoholic steatohepatitis in 29% of patients, autoimmune hepatitis in 2% of patients, primary biliary cholangitis in 1% of patients, and cryptogenic condition in 3% of patients. The proportion of patients who had nonalcoholic steatohepatitis [18,19] was higher in the arfa^®^ group than in the VIVA^®^ group. In recent studies, it was reported that the proportion of patients with liver cirrhosis caused by nonalcoholic steatohepatitis is increasing [20,21]. This report in previous studies is consistent with the findings in this study.

Functional liver reserve was lower one day after RFA than before RFA. However, the decline in functional liver reserve is not considered a clinical problem because the decline is mild. Moreover, one month after RFA, functional liver reserve reverted to the same level as before RFA. In addition, T-Bil level one day after RFA was higher in the arfa^®^ group than in the VIVA^®^ group. We posit that the difference between the groups in terms of T-Bil level one day after RFA is because tumor size is greater in the arfa^®^ group than in the VIVA^®^ group. However, this difference in T-Bil level is not considered a clinical problem because no significant difference in the Child–Pugh score was observed between the groups, and the difference in median tumor size between the groups was only 0.2 cm.

The ablation area with the 3-cm electrode needle was greater in the arfa^®^ group than in the VIVA^®^ group. However, with the 3-cm electrode needle, the difference in Dmax and Dmin between the arfa^®^ group and the VIVA^®^ group was only 0.2 cm and 0.35 cm, respectively. Further, with the 2.5-cm electrode needle, the Dmax was greater in the arfa^®^ group than in the VIVA^®^ group, and the difference in Dmax between the two groups was only 0.3 cm. The difference in ablation area between the two groups is not considered a clinical problem because the difference in Dmax and Dmin values between the two groups was only 0.2, 0.3, and 0.35 cm, respectively.

Adjustable electrode needles could not be used in previous RFA treatment procedures. While performing RFA treatment for several HCCs of different sizes, electrode needles of various sizes are required. Nonetheless, adjustable electrode needles are now available; as a result, healthcare costs can be reduced. However, switching from VIVA^®^ to arfa^®^ RFA system in October 2020 was not performed for the purpose of healthcare cost reduction. The healthcare cost is similar between the arfa^®^ and VIVA^®^ RFA systems. We adopted the arfa^®^ RFA system because its electrode needle is more lightweight and has better electrode needle recognition under the guidance of ultrasound in comparison with the VIVA^®^ RFA system. We believe that arfa^®^ will improve the performance of RFA treatment, although the reason for this notion is subjective.

There are several limitations in this study. They include the single-center design and the small sample size of the study. In addition, this study is retrospective in nature. Therefore, statistical errors are possible. The level of expertise of the hepatologist may have an effect on the diameter of the ablation area and on the change in functional liver reserve after RFA. However, an earlier study reported no difference in the performance of RFA between trainees and mentors who underwent a carefully monitored training program [22]. In our study, we could not investigate HCC recurrence and prognosis because the observation period was short. It is therefore necessary to conduct further studies to determine if there are differences in HCC recurrence and prognosis between the arfa^®^ and VIVA^®^ groups. In conclusion, there were statistically significant differences in the diameter of ablation area and in the change in functional liver reserve after RFA between patients with HCC who underwent RFA using the arfa^®^ system and those who underwent RFA using the VIVA^®^ system. However, these differences were not considered clinical problems, and hepatologists can use the arfa^®^ or VIVA^®^ RFA system according to their preference.

## Figures and Tables

**Figure 1 jcm-11-00434-f001:**
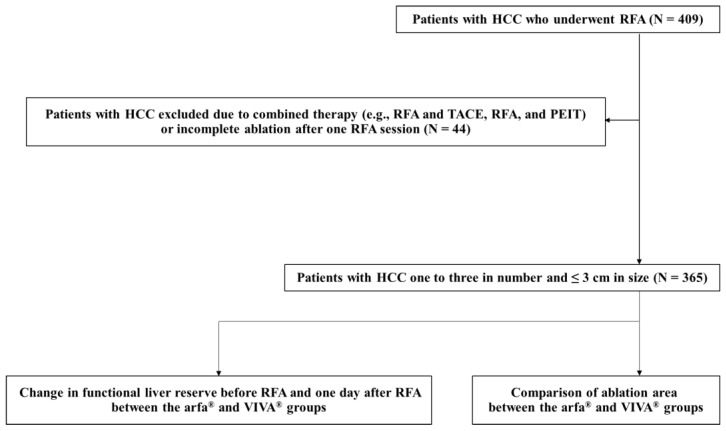
Study design. Change in functional liver reserve before RFA and one day after RFA was investigated in the arfa^®^ and VIVA^®^ groups. The ablation area in the arfa^®^ and VIVA^®^ groups was then evaluated for each electrode needle length. PEIT, percutaneous ethanol injection therapy; RFA, radiofrequency ablation; TACE, transcatheter arterial chemoembolization.

**Figure 2 jcm-11-00434-f002:**
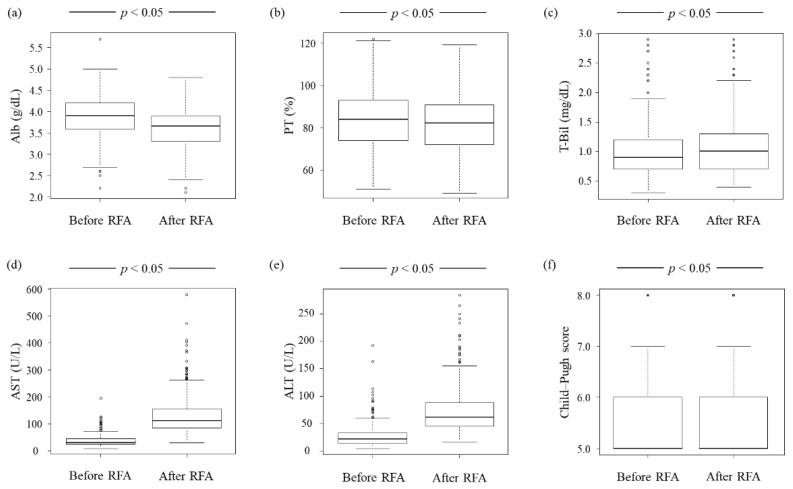
Change in functional liver reserve before RFA and 1 day after RFA. (**a**,**b**) Alb level and PT are lower 1 day after RFA than before RFA (*p* < 0.05 for both variables). (**c**–**e**) T-Bil, AST, and ALT levels are higher 1 day after RFA than before RFA (*p* < 0.05 for all three variables). (**f**) The Child–Pugh score is higher 1 day after RFA than before RFA, although it may not be apparent from figure (**f**) (*p* < 0.05). Alb, albumin; ALT, alanine aminotransferase; AST, aspartate aminotransferase; PT, prothrombin time; RFA, radiofrequency ablation; T-Bil, total bilirubin.

**Figure 3 jcm-11-00434-f003:**
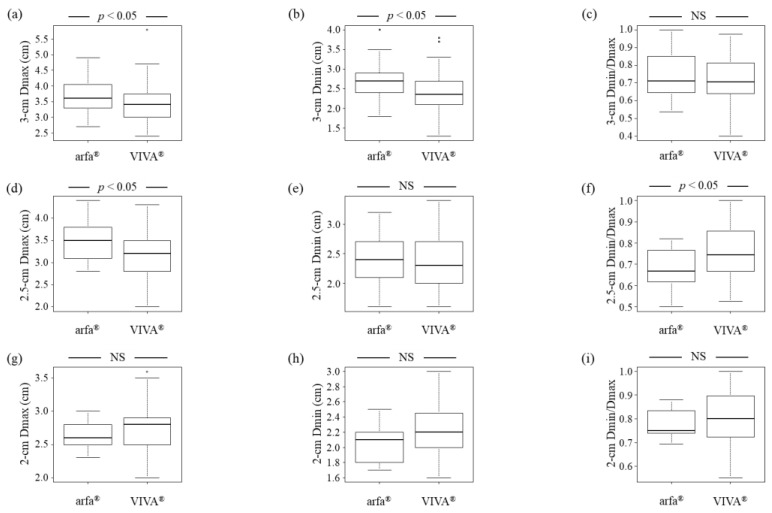
Comparison of ablation area between the arfa^®^ and VIVA^®^ groups. (**a**,**b**) With the 3-cm electrode needle, Dmax and Dmin are greater in the arfa^®^ group than in the VIVA^®^ group (*p* < 0.05 for both variables). (**c**) With the 3-cm electrode needle, no significant differences in Dmin/Dmax are observed between the arfa^®^ and VIVA^®^ groups. (**d**) With the 2.5-cm electrode needle, Dmax is greater in the arfa^®^ group than in the VIVA^®^ group (*p* < 0.05). (**e**) With the 2.5-cm electrode needle, no significant differences in Dmin are observed between the arfa^®^ and VIVA^®^ groups. (**f**) With the 2.5-cm electrode needle, Dmin/Dmax is lower in the arfa^®^ group than in the VIVA^®^ group (*p* < 0.05). (**g**–**i**) With the 2-cm electrode needle, no significant differences in Dmax, Dmin, and Dmin/Dmax are observed between the arfa^®^ and VIVA^®^ groups. Dmax, longest diameter; Dmin, shortest diameter.

**Table 1 jcm-11-00434-t001:** Characteristics of patients in the arfa^®^ and VIVA^®^ groups.

Variable	Total (*n* = 365)	arfa^®^ Group (*n* = 104)	VIVA^®^ Group (*n* = 261)	*p*-Value *
Age (years)	75 (68–80)	73 (69–80)	75 (68–80)	NS
Sex (male/female)	259/106	75/29	184/77	NS
Etiology (HBV/HCV/alcohol abuse/NASH/AIH/PBC/cryptogenic)	75/144/58/64/11/4/9	15/34/19/30 **/2/1/3	60/110/39/34/9/3/6	<0.05
Albumin (g/dL)	3.9 (3.6–4.2)	4.0 (3.6–4.3)	3.9 (3.5–4.2)	<0.05
Aspartate aminotransferase (U/L)	30 (23–43)	30 (23–40)	31 (22–44)	NS
Alanine aminotransferase (U/L)	22 (15–33)	22 (14–35)	22 (15–32)	NS
Total bilirubin (mg/dL)	0.9 (0.7–1.2)	0.9 (0.7–1.4)	0.9 (0.7–1.2)	NS
Prothrombin time (%)	84 (74–93)	91 (78–103)	82 (72–89)	<0.05
Child–Pugh score	5 (5–6)	5 (5–5)	5 (5–6)	NS
Child–Pugh class A/B	326/39	94/10	232/29	NS
Tumor size (cm)	1.7 (1.3–2.1)	1.8 (1.5–2.0)	1.6 (1.3–2.1)	<0.05
Tumor number (1/2/3)	299/59/7	92/11/1	207/48/6	NS

Data are expressed as median (interquartile range). *p*-values represent comparisons between the arfa^®^ group and the VIVA^®^ group. HBV, hepatitis B virus; HCV, hepatitis C virus; NASH, nonalcoholic steatohepatitis; AIH, autoimmune hepatitis; PBC, primary biliary cholangitis. NS, not significant. * arfa^®^ versus VIVA^®^. ** *p* < 0.05 between NASH and non-NASH versus VIVA^®^.

**Table 2 jcm-11-00434-t002:** Change in functional liver reserve between the arfa^®^ group and the VIVA^®^ group.

Variable	arfa^®^ Group (*n* = 104)	VIVA^®^ Group (*n* = 261)	*p*-Value *
Ratio of albumin one day after RFA to that before RFA	0.93 (0.88–0.98)	0.93 (0.88–0.97)	NS
Ratio of aspartate aminotransferase one day after RFA to that before RFA	3.98 (2.55–5.68)	3.52 (2.48–5.39)	NS
Ratio of alanine aminotransferase one day after RFA to that before RFA	2.81 (1.72–4.06)	2.62 (1.80–4.31)	NS
Ratio of total bilirubin one day after RFA to that before RFA	1.22 (1.00–1.43)	1.00 (0.88–1.29)	<0.05
Ratio of prothrombin time one day after RFA to that before RFA	0.98 (0.92–1.04)	0.97 (0.91–1.02)	NS
Ratio of Child–Pugh score one day after RFA to that before RFA	1.00 (1.00–1.00)	1.00 (1.00–1.14)	NS

Data are expressed as median (interquartile range). *p*-values represent comparisons between the arfa^®^ and VIVA^®^ groups. RFA, radiofrequency ablation. NS, not significant. * arfa^®^ versus VIVA^®^.

## Data Availability

Informed consent for data sharing was not obtained but the presented data are anonymized and the risk of identification is low.

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
