# Peer review of "Comparison of Ablation Area and Change in Functional Liver Reserve after Radiofrequency Ablation for Hepatocellular Carcinoma Using the arfa^®^ and VIVA^®^ Systems"

_jcm, 2022, doi:10.3390/jcm11020434_

Round 1
Reviewer 1 Report
The similar comparison of resection systems in HCC was not shown so far. Since there are no significant differences, in evidence based medicine the result of this study can be valuable for patients and for specialists. Especially in dispute situations or occurrence of complications the knowledge of outcome of surgical treatment, that can be cited, is valuable. In this study patients were well defined, both pre-operative and post operative.
Author Response
We thank you for your careful review of our manuscript and for your useful comments, which served as the basis for revising our manuscript. In addition, we sought English proofreading again. Our responses to your comments are as follows:
Response to Reviewer 1
Thank you for valuable comment and careful review of our manuscript.
Reviewer 2 Report
This is an interesting area of research. New RFA systems came up every now and then. It is important to keep up to date and add new technology for the patients sake.
Introduction:
Line 30-31: HCC is mainly caused by HCV-induced cirrhosis. However, other causes include ....
Line 32-34: To be removed (Reference 8) as it does not needed in this area.
Line 49-51 (Aim of the study): to be changed to: To compare between both arfa and VIVA RFA systems in patients with HCC as regards to both ablation area and functional liver reserve.
Materials and Methods:
Line 59-63: exclusion criteria includes patients with .........
Line 75-77: to be removed as it is repeated.
Line 78-79: to be removed and add a title "abbreviations" after the abstract where you can add all abbreviations you need.
Results:
Table (1): Variable etiology: It is corrupted and need to be cleared in the table Then, I do not understand what you mean by significance in this area???!! If you mean increase number of NASH, you had to put asterixis above it in the table. Again what you means by other causes?? you better mention the causes.
Change of functional liver reserve:
Line 128-138: better to rephrase to: The absolute value of all liver parameters showed statistically significant changes before and one day after RFA. However, Only total bilirubin ratio before and one day after RFA was statistically significant.
Comparison of ablation area:
Line 160-162 from "Since only a few.... to in these HCCs" to be transferred to line 156 before the sentence : we determined..".
Discussion:
Line 202-204: to be removed.
Line 220: Dmaxs and Dmins.
Study limitations: you have to mention it is a retrospective study that may add statistical errors.
You need to add a paragraph discussing the economic aspect and costs of both systems. Also, you need to add a sentence declaring why you change your ablation system in Japan to another system?? Is it a cost manner or new modifications that give the patients privileges?
Author Response
We thank you for your careful review of our manuscript and for your useful comments, which served as the basis for revising our manuscript. In addition, we sought English proofreading again. Our responses to the each of your comments are as follows:
Response to Reviewer 2
Thank you for reviewing our manuscript. Our responses to your comments are as follows:
1) Line 30-31: HCC is mainly caused by HCV-induced cirrhosis. However, other causes include ....
Response:
Thank you for your valuable comment. We changed “HCC is caused by liver cirrhosis, and hepatitis C virus is one of the major causes of liver cirrhosis” to “HCC is mainly caused by hepatitis C virus-induced liver cirrhosis. However, other causes include hepatitis B virus, excessive alcohol consumption, nonalcoholic steatohepatitis, autoimmune hepatitis, and primary biliary cholangitis.”
2)Line 32-34: To be removed (Reference 8) as it does not needed in this area.
Response:
In line with your comment, we removed the following statement: “In Japan, patients with hepatitis C are usually advanced in age. This characteristic is explained by the widespread transmission of hepatitis C infection in Japan in the 1930s; in contrast, hepatitis C infection was extensively spread in the United States in the 1960s.”
3)Line 49-51 (Aim of the study): to be changed to: To compare between both arfa and VIVA RFA systems in patients with HCC as regards to both ablation area and functional liver reserve.
Response:
Thank you for your valuable comment. We changed “We investigated the change in functional liver reserve before RFA and one day after RFA. We also compared the ablation area to evaluate the performances of the arfa® and VIVA® RFA systems in patients with HCC,” to “In this study, we compared arfa® and VIVA® RFA systems with respect to the ablation area and functional liver reserve in patients with HCC.
4)Line 59-63: exclusion criteria includes patients with .........
Response:
According to your comment, we changed “We excluded patients with…” to “The exclusion criterion was…” and “We also excluded patients with…” to “The other exclusion criterion was…”.
5)Line 75-77: to be removed as it is repeated.
6)Line 78-79: to be removed and add a title "abbreviations" after the abstract where you can add all abbreviations you need.
Response:
Thank you for your valuable comment. However, we retained these sentences because they contain explanations to Figure 1.
7)Table (1): Variable etiology: It is corrupted and need to be cleared in the table Then, I do not understand what you mean by significance in this area???!! If you mean increase number of NASH, you had to put asterixis above it in the table. Again what you means by other causes?? you better mention the causes.
Response:
Thank you for your insightful comment. We did not revise Table 1 because the publisher will revise it for us. Fisher’s exact test was used in analyzing the p-values of all etiologies between the arfa® and VIVA® groups and those between NASH and non-NASH. In addition, we changed “others” to “cryptogenic.”
8)Line 128-138: better to rephrase to: The absolute value of all liver parameters showed statistically significant changes before and one day after RFA. However, Only total bilirubin ratio before and one day after RFA was statistically significant.
Response:
We appreciate your comment. We changed the following text as suggested:
“Alb level and PT were lower one day after RFA than before RFA (P < 0.05 for both variables) (Figures 2a and 2b). In contrast, total bilirubin (T-Bil), aspartate aminotransferase (AST), and alanine aminotransferase (ALT) levels were higher one day after RFA than before RFA (P < 0.05 for all three variables) (Figures 2c, 2d, and 2e). Although it may not be apparent from Figure 2f, Child–Pugh score was significantly higher one day after RFA than before RFA (P < 0.05). Further, we compared change in functional liver reserve between the arfa® and VIVA® groups. The ratio of T-Bil level one day after RFA to that before RFA was higher in the arfa® group than in the VIVA® group (P < 0.05). In contrast, there were no differences in the ratios of Alb level, AST level, ALT level, PT, and Child–Pugh score one day after RFA to those before RFA between the arfa® and VIVA® groups (Table 2).”
To
“The absolute value of all liver parameters showed statistically significant changes before and 1 day after RFA (Figure 2). Between the arfa® and VIVA® groups, only the ratio of total bilirubin (T-Bil) level 1 day after RFA to that before RFA was statistically significant (Table 2).”
9)Line 160-162 from "Since only a few.... to in these HCCs" to be transferred to line 156 before the sentence : we determined..".
Response:
Thank you for your valuable comment. We transferred “Considering that only a few HCCs were treated using a 1.5- or 1-cm 17G adjustable electrode needle, we did not evaluate the ablation area in these HCCs.” Next to this sentence is the following: “We determined the Dmax and Dmin of the ablation area for 115 HCCs….”
10) Line 202-204: to be removed.
Response:
In line with your comment, we removed “Furthermore, Alb level and PT before RFA were higher in the arfa® group than in the VIVA® group. Recently, there has been progress in the treatment of liver cirrhosis every year. The difference in Alb level and PT before RFA between the two groups may influence progress in the treatment of liver cirrhosis”.
11)Line 220: Dmaxs and Dmins.
Response:
According to your comment, we changed “Dmax and Dmin” to “Dmax and Dmin values.”
12)Study limitations: you have to mention it is a retrospective study that may add statistical errors.
Response:
Thank you for this important comment. Accordingly, the following phrase has been added: “In addition, this study is retrospective in nature. Therefore, statistical errors are possible.”
13)You need to add a paragraph discussing the economic aspect and costs of both systems. Also, you need to add a sentence declaring why you change your ablation system in Japan to another system?? Is it a cost manner or new modifications that give the patients privileges?
Response:
In line with your comment, we added the following paragraph:
“Adjustable electrode needles could not be used in previous RFA treatment procedures. While performing RFA treatment for several HCCs of different sizes, electrode needles of various sizes are required. Nonetheless, adjustable electrode needles are now available; as a result, healthcare costs can be reduced. However, switching from VIVA® to arfa® RFA system in October 2020 was not performed for the purpose of healthcare cost reduction. The healthcare cost is similar between the arfa® and VIVA® RFA systems. We adopted the arfa® RFA system because its electrode needle is more lightweight and has better electrode needle recognition under the guidance of ultrasound in comparison with the VIVA® RFA system. We believe that arfa® will improve the performance of RFA treatment, although the reason for this notion is subjective.”